# Atypical Presentation of Enlarged Vestibular Aqueducts Caused by *SLC26A4* Variants

**DOI:** 10.3390/children9020165

**Published:** 2022-01-28

**Authors:** Jun Chul Byun, Kyu-Yup Lee, Su-Kyeong Hwang

**Affiliations:** 1Department of Pediatrics, School of Medicine, Keimyung University Dongsan Medical Center, Daegu 42601, Korea; jcbyun@dsmc.or.kr; 2Department of Otorhinolaryngology-Head and Neck Surgery, School of Medicine, Kyungpook National University, Daegu 41944, Korea; kylee@knu.ac.kr; 3Department of Pediatrics, School of Medicine, Kyungpook National University, Daegu 41944, Korea

**Keywords:** Horner syndrome, ptosis, anisocoria, deafness, vestibular aqueduct, pendrin, *SLC26A4*

## Abstract

Enlarged vestibular aqueduct is the most common inner ear malformation in pediatric patients with sensorineural hearing loss. Here, we report a new presentation of enlarged vestibular aqueduct in a Korean family. The family consists of two parents and five daughters, and the first and second daughters were diagnosed with bilateral enlarged vestibular aqueducts. The third daughter, who showed no signs of hearing deterioration, came to medical attention with incomplete Horner syndrome. Evaluations for localization of Horner syndrome on the patient and Sanger sequencing of *SLC26A4* on the family members were performed. Although auditory brainstem response and pure tone audiometry of the third daughter were normal, temporal bone computed tomography demonstrated bilateral enlarged vestibular aqueducts. Sanger sequencing of *SLC26A4* revealed compound heterozygous variants c.2168A>G and c.919-2A>G in the first, second, and third daughters. Diagnosis of enlarged vestibular aqueduct is often delayed because the degree of hearing loss can vary, and a considerable phenotypic variability can be shown even in family members with the same *SLC26A4* variations. Fluctuations of CSF pressure into the cochlear duct and recurrent microruptures of the endolymphatic membrane could result in damage of sympathetic nerve supplying to the inner ear, which could explain the mechanism of Horner syndrome associated with enlarged vestibular aqueduct.

## 1. Introduction

Hearing loss is one of the most common communication disorders in humans. Approximately 1 in 1000 children are born with hearing loss [1], and the recent World Health Organization (WHO) estimate suggests that approximately 6.1% of the world’s population were living with disabling hearing loss [2]. Sensorineural hearing loss (SNHL) is associated with abnormalities of inner ear structures. Approximately 80% of these cases are affected by pathogenic variants in genes related to the hearing process [3], with *SLC26A4* being one of the major contributors to hereditary hearing loss [4,5,6]. While *GJB2* is the most prevalent cause of hereditary hearing loss in western populations [7,8], *SLC26A4* is the most prevalent in eastern Asian people; that is, 82% in Japanese people [9], 97.9% in Chinese people [10], and 92% in Korean people [11]. *SLC26A4* is on chromosome 7q22.3 and encodes a transmembrane ion transporter called pendrin, which a multifunctional anion exchanger that has an affinity to chloride, iodide, bicarbonate, and other anions [12]. Pendrin is expressed in multiple organs. Pendrin expressed in the cochlea and vestibule of inner ear plays an important role in endolymphatic fluid resorption, acid–base balance, and proper function of the inner ear [13]. Pendrin expressed in thyrocyte plays as a chloride/iodide exchanger that is essential for cellular iodide efflux into the follicular lumen, and a dysfunction of pendrin usually leads to the partial impairment of thyroid organification [13]. Pendrin located at or near cortical collecting ducts of kidneys acts as a chloride/anion exchanger [12]; thus, it plays an important role in the regulation of blood pressure and fluid balance [13,14], and defects in pendrin can cause metabolic alkalosis [15]. Pendrin expressed in bronchial epithelial cells helps to regulate airway surface liquid thickness via its function as a chloride/bicarbonate exchanger. The defective pendrin may lead to excessive production of airway mucus, which is a cardinal feature of bronchial asthma and chronic obstructive pulmonary disease (COPD) [16]. Pathogenic variants of *SLC26A4* are associated with Pendred syndrome and nonsyndromic cases of SNHL, depending on the presence of thyroid phenotype. Thyroid phenotype in Pendred syndrome is usually a euthyroid goiter but can be variable, ranging from thyroid goiter with hypothyroidism to thyroid cancer. Both Pendred syndrome and nonsyndromic SNHL demonstrate a clinical characteristic of the inner ear, which radiologically manifests as enlarged vestibular aqueduct (EVA) [17]. The classic radiographic description is an enlargement of the vestibular aqueduct >1.5 mm in diameter, and the newer Cincinnati criteria define EVA as a diameter ≥1.0 mm at the midpoint or 2.0 mm at the operculum [18]. Here, we report a new presentation of EVA in a Korean family.

## 2. Materials and Methods

Detailed clinical and family histories were obtained on the family members with EVA. The institutional review board of Kyungpook National University Hospital approved the protocol, and informed consent was obtained for the genetic analysis and the use of the results for diagnosis and research purposes from the patient’s legal guardian (IRB no. KNUH 2016-06-011). Genomic DNA was extracted from ethylenediaminetetraacetic acid (EDTA)-treated whole blood samples using a QIAamp DNA Blood Mini kit (ID: 51106; Qiagen, Hilden, Germany). DNA quality and quantity were assessed using a Qubit Fluorometer (Invitrogen, Carlsbad, CA, USA) and a Quant-iT BR assay kit (Q32850; Invitrogen, Carlsbad, CA, USA). Sanger sequencing of *SLC26A4* was performed on the family members. All coding exons and flanking intron sequences of *SLC26A4* were amplified with forward and reverse primers. The amplified products were directly sequenced using an automated DNA sequencer (ABI3130; Applied Biosystems, Foster City, CA, USA) using a Big-Dye Terminator Cycle Sequencing kit version 3.1. The primer sequences are available upon request. Evaluations included brain magnetic resonance imaging, computed tomography (CT) of temporal bone, neck, and chest, cerebrospinal fluid analysis and urinary catecholamines, thyroid function test, auditory brainstem response (ABR), and pure tone audiometry (PTA).

## 3. Results

The family consists of parents and five daughters, and the first and second daughters were diagnosed with SNHL (Figure 1). The parents and the other daughters had no hearing impairment. The first daughter showed normal development, including language development by the age of 3. She was born full-term by normal delivery, not requiring special care baby unit, and passed her newborn hearing screening. At the age of 4, she visited the hospital because her pronunciation became increasingly awkward, she could not understand other people’s words well, and she was often seen turning up the volume of television. The results of ABR revealed identifiable wave V at 60 dB nHL in both ears. Temporal CT showed bilateral EVA. She has used hearing aids since the age of 5. She has shown recurrent paroxysmal vertigo and has been hospitalized several times due to fluctuating hearing loss after minor head trauma.

The second daughter visited the hospital due to delayed language development at the age of 2. She did not use any words except “mommy” or “daddy”, respond when spoken to, or react to loud noises. She was born prematurely at 36 weeks and had a birthweight of 2900 g. The prenatal care and delivery were uneventful, but the patient was admitted to the neonatal intensive care unit for respiratory distress and was treated for three weeks. She passed her newborn hearing screening, but later became less responsive to sound. Her ABR showed identifiable wave V at 90 dB nHL in right ear and no response to 90 dB nHL in left ear. Temporal CT showed bilateral EVA. She received cochlear implants and had a few days of dizziness during the hospitalization but had no vestibular symptoms afterward unlike the first daughter. Sanger sequencing of SLC26A4 on the first and the second daughters revealed compound heterozygous variants c.2168A>G and c.919-2A>G. While c.2168A>G causes p.His723R, c.919-2A>G is a splice acceptor variant. Sanger sequencing of the SLC26A4 variants on the parents revealed that p.His723Arg was inherited from the father and c.919-2A>G was inherited from the mother. The third daughter first came to medical attention at 4 years of age due to sudden onset gait difficulties. She showed a head tilt to the left side (Figure 2A), horizontal nystagmus, and ataxic gait. From the day before admission, she had complained of dizziness and vomited several times. Upon neurological examination, muscle power, muscle tone, sensation, deep tendon reflexes, and cerebellar function test were normal. An examination of the eyes revealed ptosis of right upper eyelid, a round and constricted right pupil that exhibited dilation lag. Anhidrosis was assessed using a metal spoon and showed normal perspiration. Her neurological examination suggested incomplete Horner syndrome consisting of right ptosis, right miosis, but not anhidrosis. Imaging of the brain, neck, and chest, cerebrospinal fluid analysis, and urinary catecholamines for localization of Horner syndrome showed no abnormalities. Although the third daughter passed her newborn hearing screening and showed no signs of hearing deterioration in daily life, temporal bone CT and ABR were conducted because she had a family history of EVA, and there were no abnormal findings in the other tests. The temporal bone CT examinations of the patient showed bilateral EVA (right 1.95 mm, left 2.20 mm) (Figure 2B). Laboratory tests, including a thyroid function test, were normal, and thyroid goiter was not detected. The same compound, heterozygous *SLC26A4*, with variants with her older sisters were identified in the patient. The ABR and PTA were normal during the hospitalization, but hearing loss has gradually progressed, and the ABR implemented 9 months later showed identifiable wave V at 30 dB nHL in right ear and 60 dB nHL in left ear (Figure 2C). The patient had two more recurrences of Horner syndrome after discharge.

The fourth and fifth daughters who are dizygotic twins did not have any symptoms related to EVA including hearing impairment. The twins revealed heterozygous c.919-2A>G only in Sanger sequencing of the *SLC26A4* variants.

## 4. Discussion

EVA is the most common inner ear malformation, affecting up to 15% of pediatric patients with SNHL [19]. The variant types of *SLC26A4* causing EVA are variable, including missense/nonsense single nucleotide variations (SNVs), splicing variations, small indels, and gross indels. So far, 587 relevant variations have been reported in HGMD (Human Gene Mutation Database, professional). There are population-specific differences in the allele-specific variation rates. In northern European populations, p.Leu236Pro, p.Thr416Pro, c.1001+1G>A, and p.Glu384Gly are seen more frequently than other pathogenic variants in patients with Pendred syndrome and nonsyndromic EVA [20,21]. In Chinese, Japanese, Korean, and Pakistani populations, c.919-2A>G, p.His723Arg, and p.Val239Asp are prevalent pathogenic variants [11,22,23,24,25].

Diagnosis of EVA is often delayed because the degree of hearing loss can vary from normal to profound at the time of diagnosis and the hearing loss can be fluctuant in patients with EVA. In addition to deficits in hearing, individuals with EVA may experience vestibular symptoms or various phenotypic spectrum because pendrin is a multifaceted transporter expressed in multiple organs. The mechanism of hearing loss and vertigo in EVA is not clear. One proposed mechanism is that intracranial pressure changes results in inner ear pressure changes by a widened vestibular aqueduct [26,27]. Like Ménière’s disease, periodic microruptures in the already distended endolymphatic membranes leads to disturbances in ion concentration in the endolymphatic and perilymphatic spaces, loss of the membrane potential, and resultant alterations in hearing and vestibular function [26,28]. Due to the altered fluid flow of the inner ear system, worsening hearing thresholds have been associated with head trauma, strenuous exercise, and upper respiratory infections [28]. A recent meta-analysis revealed that decreased hearing after minor head trauma was reported in one-third of patients with EVA [29].

Horner syndrome is caused by damage to a certain pathway in the sympathetic nervous system. The postganglionic (third-order) sympathetic neurons originated in the superior cervical ganglion often results in Horner’s syndrome, which travel in the wall of the internal carotid artery and continue on to the cavernous sinus [30]. Lesions proximal to the superior cervical ganglion will have anhidrosis of the entire ipsilateral head and neck. In contrast, lesions distal to the superior cervical ganglion may not show anhidrosis. Enlargement of the endolymphatic sac not only leads to cochlear and vestibular dysfunction but could cause recurrent Horner syndrome, as in our patient’s case. The inner ear receives input from the trigeminal and sympathetic nerve through the tympanic plexus. The superior cervical ganglion provides sympathetic innervation to vestibular structures including the neuroepithelium of the semicircular canals and otolith organs, providing a conceivable substrate for modulation of vestibulo-sympathetic reflexes. A neurologically based hypothesis is that sudden fluctuations in CSF pressure with highly concentrated proteins into the cochlear duct and recurrent microruptures of the endolymphatic membrane could result in damage of sympathetic nerve supplying to the inner ear. Consequently, a disruption of sympathetic activity could result in Horner syndrome associated with EVA. Further research is needed to clarify the association between Horner syndrome and EVA.

This study broadens the phenotypic characteristics and expands our understanding of EVA. Diagnosis of EVA in patients with Horner syndrome requires a high index of suspicion. Clinical events that should suggest EVA include (1) SNHL has previously been diagnosed in the patient or in his/her family members, (2) vestibular symptoms such as ataxia, head tilt, or nystagmus are combined, and/or (3) typical evaluations for Horner syndrome including imaging of brain, neck, and chest, and analysis of cerebrospinal fluid and urinary catecholamines show no abnormalities. When there is any suspicion suggesting EVA, temporal CT and genetic study should be considered. In summary, EVA can demonstrate a considerable variability in phenotypic expression even in family members with the same *SLC26A4* variations. Genotype–phenotype correlations have not yet been found in EVA. The repeated symptoms of Horner syndrome also suggest the association with EVA. Careful history taking and neurologic examination may provide the first clues to an early diagnosis of EVA.

## Figures and Tables

**Figure 1 children-09-00165-f001:**
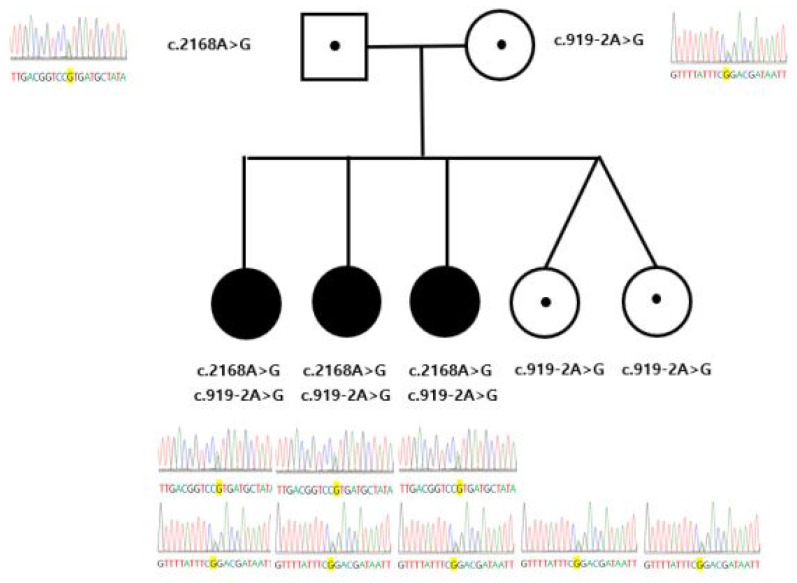
Pedigree of the family and sequencing chromatograms of each family members.

**Figure 2 children-09-00165-f002:**
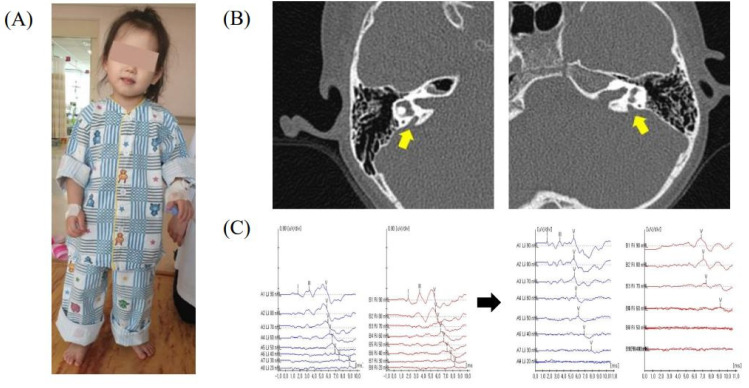
(**A**) General appearance of the third daughter at the time of visit. (**B**) The temporal bone computed tomography of the third daughter showing bilateral enlarged vestibular aqueduct. (**C**) The ABR data during the hospitalization and after 9 months.

## Data Availability

Data presented in this study are available from the corresponding author upon reasonable request.

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
