# Peer review of "Atypical Presentation of Enlarged Vestibular Aqueducts Caused by SLC26A4 Variants"

_children, 2022, doi:10.3390/children9020165_

Round 1
Reviewer 1 Report
I read the manuscript which were well written of feature of enlarged vestibular aqueduct that follows autosomal recessive inheritance manner.  I have some suggestions of miner revisions of the manuscript.
1, In figure 1, Add to ABR data (indicating I to V waves) of third daughter.
2, In figure 1, Add to each sangar sequence data of families under the Family tree.
3, Put together the Fig.2 and Fig.3.
4, In Fig.3, Add to arrows indicating enlarged vestibular aqueduct on CT figure.
Author Response
- In figure 1, Add to ABR data (indicating I to V waves) of third daughter.
- The ABR data was added to figure 2, because figure 2 includes the third daughter.
- In figure 1, Add to each sangar sequence data of families under the Family tree.
- Each Sanger sequencing data was added to the family tree.
- Put together the Fig.2 and Fig.3.
- Figure 1 and figure 2 were put together.
- In Fig.3, Add to arrows indicating enlarged vestibular aqueduct on CT figure.
- Arrows were added to figure 3 indicating EVA.
Reviewer 2 Report
Major point:
If any relationship between EVA and Horner syndrome existed, authors should refer many publications pointing to it. If no such publications existed, authors should be more cautious about interpreting their results. Authors should discuss why only the third daughter had the Horner syndrome although the first and the second daughters had the same compound heterozygous variants as the third one.
Minor points:
- Did patient(s) suffer from thyroid goiter?
- Authors should make clear that c.2168A>G causes p.H723R while c.919-2A>G causes no change in the amino acid.
Author Response
<Major poin answer>
Since this study is the first report to reveal the association of EVA and Horner syndrome, there is no publication of similar cases. As mentioned in the manuscript, EVA can exhibit various clinical aspects even in family members with the same mutation. It was additionally mentioned that genotype–phenotype correlations have not yet been found in EVA, and the patient had two more recurrences of Horner syndrome after discharge. The repeated symptoms of Horner syndrome also suggest the association with EVA.
<Minor point answer>
Minor points:
- Did patient(s) suffer from thyroid goiter?
- The patient did not suffer from thyroid goiter, and it was added to the manuscript.
2. Authors should make clear that c.2168A>G causes p.H723R while c.919-2A>G causes no change in the amino acid.
- While 2168A>G causes p.H723R, c.919-2A>G is a splice acceptor variant, and it was added to the manuscript.

Round 2
Reviewer 2 Report
I found authors revised the manuscript in addressing all questions raised in the review process.
Author Response
We appreciate for the review. thanks to your comments, the manuscript became much better and resonable